# Optimal Randomness for Stochastic Configuration Network (SCN) with Heavy-Tailed Distributions

**DOI:** 10.3390/e23010056

**Published:** 2020-12-31

**Authors:** Haoyu Niu, Jiamin Wei, YangQuan Chen

**Affiliations:** 1Electrical Engineering and Computer Science Department, University of California, Merced, CA 95340, USA; hniu2@ucmerced.edu; 2School of Telecommunications Engineering, Xidian University, No.2, Taibai Road, Xi’an 710071, Shaanxi, China; jmwei@xidian.edu.cn

**Keywords:** SCN, optimal randomness, heavy-tailed distribution, Lévy, Weibull, Cauchy

## Abstract

Stochastic Configuration Network (SCN) has a powerful capability for regression and classification analysis. Traditionally, it is quite challenging to correctly determine an appropriate architecture for a neural network so that the trained model can achieve excellent performance for both learning and generalization. Compared with the known randomized learning algorithms for single hidden layer feed-forward neural networks, such as Randomized Radial Basis Function (RBF) Networks and Random Vector Functional-link (RVFL), the SCN randomly assigns the input weights and biases of the hidden nodes in a supervisory mechanism. Since the parameters in the hidden layers are randomly generated in uniform distribution, hypothetically, there is optimal randomness. Heavy-tailed distribution has shown optimal randomness in an unknown environment for finding some targets. Therefore, in this research, the authors used heavy-tailed distributions to randomly initialize weights and biases to see if the new SCN models can achieve better performance than the original SCN. Heavy-tailed distributions, such as Lévy distribution, Cauchy distribution, and Weibull distribution, have been used. Since some mixed distributions show heavy-tailed properties, the mixed Gaussian and Laplace distributions were also studied in this research work. Experimental results showed improved performance for SCN with heavy-tailed distributions. For the regression model, SCN-Lévy, SCN-Mixture, SCN-Cauchy, and SCN-Weibull used less hidden nodes to achieve similar performance with SCN. For the classification model, SCN-Mixture, SCN-Lévy, and SCN-Cauchy have higher test accuracy of 91.5%, 91.7% and 92.4%, respectively. Both are higher than the test accuracy of the original SCN.

## 1. Introduction

The Stochastic Configuration Network (SCN) model is generated incrementally by using stochastic configuration (SC) algorithms [1]. Compared with the existing randomized learning algorithms for single-layer feed-forward neural networks (SLFNNs), the SCN can randomly assign the input weights (w) and biases (b) of the hidden nodes in a supervisory mechanism, which is selecting the random parameters with an inequality constraint and assigning the scope of the random parameters adaptively. It can ensure that the built randomized learner models have universal approximation property. Then, the output weights are analytically evaluated in either a constructive or selective manner [1]. In contrast with the known randomized learning algorithms, such as the Randomized Radial Basis Function (RBF) Networks [2] and the Random Vector Functional-link (RVFL) [3], SCN can provide good generalization performance at a faster speed. Concretely, there are three types of SCN algorithms, which are SC-I, SC-II, and SC-III. SC-I algorithm uses a constructive scheme to evaluate the output weights only for the newly added hidden node [4]. All of the previously obtained output weights are kept the same. The SC-II algorithm recalculates part of the current output weights by analyzing a local least squares problem with user-defined shifting window size. The SC-III algorithm finds all the output weights together by solving a global least-squares problem.

SCN algorithms have been commonly studied and used in many areas, such as image data analytics [5,6], prediction of component concentrations in sodium aluminate liquor [7], and etc. [8,9]. For example, in [5], Li et al. developed a two-dimensional SCNs (2DSCNs) for image data modelling tasks. Experimental results on hand written digit classification and face recognition showed that the 2DSCNs have great potential for image data analytics. In [7], Wang et al. proposed a SCN-based model for measuring component concentrations in sodium aluminate liquor, which are usually acquired by titration analysis and suffered from larger time delays. From the results, the mechanism model showed the internal relationship. The improved performance can be achieved by using the SCN-based compensation model. In [10], Lu et al. proposed a novel robust SCN model based on a mixture of the Gaussian and Laplace distributions (MoGL-SCN) in the Bayesian framework. To improve the robustness of the SCN model, the random noise of the SCN model is assumed to follow a mixture of Gaussian distribution and Laplace distributions. Based on the research results, the proposed MOGL-SCN could construct prediction intervals with higher reliability and prediction accuracy.

Neural Networks (NNs) can learn from data to train feature-based predictive models. However, the learning process can be time-consuming and infeasible for applications with data streams. An optimal method is to randomly assign the weights of the NNs so that the task can become a linear least-squares problem. In [11], Wang et al. classified the NN models into three types. First, the feed-forward networks with random weights (RW-FNN) [12]. Second, recurrent NNs with random weights [13]. Third, randomized kernel approximations [14]. According to [11], there are three benefits of the randomness: (1) simplicity of implementation, (2) faster learning and less human intervention, (3) possibility of leveraging linear regression and classification algorithms. Randomness is used to define a feature map, which converts the data input into a high dimensional space where learning is more simpler. The resulting optimization problem becomes a standard linear least-squares, which is a simpler and scalable learning procedure.

For the original SCN algorithms, weights and biases are randomly generated in uniform distribution. Randomness plays a significant role in both exploration and exploitation. A good NNs architecture with randomly assigned weights can easily outperform a more deficient architecture with finely tuned weights [11]. Therefore, it is critical to discuss the optimal randomness for the weights and biases in SCN algorithms. In this study, the authors mainly discussed the impact of three different heavy-tailed distributions on the performance of the SCN algorithms, Lévy distribution, Cauchy distribution, and Weibull distribution [15]. Heavy-tailed distribution has shown optimal randomness for finding targets [16], which plays a significant role in exploration and exploitation [17]. It is important to point out that the proposed SCN models are very different from Lu et al. [10]. As mentioned earlier, Lu at al. assumed that the random noise of the SCN model following a mixture of Gaussian distribution and Laplace distributions. In this research study, the authors randomly initialize the weights and biases with heavy-tailed distributions instead of uniform distribution. To compare with the mixture distributions, the authors also used the mixture distributions for weight and bias generation. A more detailed comparison of the two heavy-tailed methods is shown in Results and Discussion section.

There are two objectives for this research, (1) compare the performance of SCN algorithms with heavy-tailed distributions on a linear regression model [18]; (2) evaluate the SCN algorithms performance on MNIST handwritten digit classification problem with heavy-tailed distributions. The rest of the paper is organized as follows: Section 2 introduces fundamental definitions of the heavy-tailed distributions and how to generate the random numbers according to the given distribution. Results and discussions are presented in Section 3. A simple regression model and MNIST handwritten digit classification problem are used to demonstrate the heavy-tailed SCNs. In Section 4, the authors draw conclusive remarks and share views in SCN with heavy-tailed distributions in future research.

## 2. Materials and Methods

### 2.1. Heavy-Tailed Distributions

Heavy-tailed distributions are widely used for modeling in different scenarios, such as finance [19], insurance [20], and medicine [21]. The distribution of a real-valued random variable X is said to have a heavy right tail if the tail probabilities P(X>x) decay more slowly than those of any exponential distribution if
(1)$$\lim_{x\to \infty }\left(\frac{P(X>x)}{e^{-\lambda x}}\right)=\infty ,$$
for every λ>0. For the heavy left tail, it has similar definition [22].

#### 2.1.1. Lévy Distribution

Lévy distribution, named after a French mathematician Paul Lévy, is a random walk that has a heavy-tailed distribution [23]. As a fractional-order stochastic process with heavy-tailed distributions, Lévy distribution has better computational effects [24]. Lévy distribution is stable and has probability density functions that can be expressed analytically. The probability density function of Lévy flight [25] is
(2)$$p\left(x,\mu ,\gamma \right)=\left\{\begin{array}{cc} \frac{\sqrt{\frac{\gamma }{2\pi }}}{e^{\frac{\gamma }{2(x-\mu )}{\left(x-\mu \right)}^{3/2}}}, & x>\mu ,\\ 0, & x\leq \mu . \end{array}\right.$$
where μ is the location parameter and γ is the scale parameter. In practice, the Lévy distribution is updated by
(3)$$Lévy\left(\beta \right)=\frac{u}{{\left|\nu \right|}^{1/\beta }},$$
where *u* and ν are random numbers generated from a normal distribution with mean of 0 and standard deviation of 1. The stability index β ranges from 0 to 2. Moreover, it is interesting to point out that the well-known Gaussian and Cauchy distribution are its special cases when its stability index is set to 2 and 1, respectively.

#### 2.1.2. Weibull Distribution

A random variable is subject to a Weibull distribution if it has a tail function *F* as
(4)$$F\left(x\right)=e^{-{\left(x/k\right)}^{\lambda_{w}}},$$
where k>0 is the scale parameter, $\lambda_{w}>0$ is the shape parameter. If the $\lambda_{w}<1$, the Weibull distribution will be a heavy-tailed distribution.

#### 2.1.3. Cauchy Distribution

A random variable is subject to the Cauchy distribution if its cumulative distribution is:(5)$$F\left(x\right)=\frac{1}{\pi }arctan\left(\frac{2(x-\mu_{c})}{\sigma }\right)+\frac{1}{2},$$
where $\mu_{c}$ is the location parameter, σ is the scale parameter.

### 2.2. Mixture Distributions

A mixture distribution is derived from a collection of other random variables. First, a random variable is selected by chance from the collection according to given probabilities of selection. Then, the value of the selected random variable is realized. The mixture distribution densities are complicated in terms of simpler densities, which provide a good model for certain data sets. The different subsets of the data can exhibit different characteristics. Therefore, the mixed distributions can effectively characterize the complex distributions of certain real-world datasets. In [10], a robust SCN based on the mixture of the Gaussian and Laplace distributions were proposed. Thus, Gaussian and Laplace distributions are mentioned in this section for comparison purpose.

#### 2.2.1. Gaussian Distribution

A random variable X has a Gaussian distribution with mean $\mu_{G}$ and variance $\sigma_{G}^{2}$ (−∞ < $\mu_{G}$ < *∞* and $\sigma_{G}$ > 0) if X has a continuous distribution for which the probability density function is as follows [26]:(6)$$f\left(x|\mu_{G},\sigma_{G}^{2}\right)=\frac{1}{{\left(2\pi \right)}^{1/2}\sigma_{G}}e^{-\frac{1}{2}{\left(\frac{x-\mu_{G}}{\sigma_{G}}\right)}^{2}},for-\infty <x<\infty .$$

#### 2.2.2. Laplace Distribution

The probability density function of the Laplace distribution can be written as follows [10]:(7)$$F\left(x|\mu_{l},\eta \right)=\frac{1}{{\left(2\eta^{2}\right)}^{1/2}}e^{\left(-\frac{\sqrt{2}\left|x-\mu_{l}\right|}{\eta }\right)},$$
where $\mu_{l}$ and η represent the location and scale parameters, respectively.

### 2.3. SCN with Heavy-Tailed Distribution

SCN was first proposed by Wang et al. in 2017 [1]. Compared with the known randomized learning algorithms for single hidden layer feed-forward neural networks, the SCN randomly assigns the input weights and biases of the hidden nodes in the light of a supervisory mechanism. The output weights are analytically evaluated in a constructive or selective method. The SCN has better performance than other randomized neural networks in terms of the fast learning, scope of the random parameters, and the required human intervention. Therefore, it has already been used in many data processing projects, such as [5,9,27].

Since the parameters of hidden layers are randomly generated in uniform distribution, it might exist optimal randomness in the SCN model. Heavy-tailed distribution has shown optimal randomness in an unknown environment for finding some targets [17], which plays a significant role in both exploration and exploitation. Therefore, in this research, the authors used heavy-tail distributions to randomly update the weights and biases in the hidden layers to see if the SCN models have improved performance with heavy-tailed distributions. Some of the key parameters of the SCN are listed in Table 1. For example, the maximum times of random configuration T${}_{max}$ is set as 200. The scale factor Lambdas in the activation function, which directly determines the range for the random parameters, is examined by performing different settings (0.5–200). The tolerance is set as 0.05. Most of the parameters for SCN with heavy-tailed distributions are kept the same with the original SCN algorithms for comparison purpose. For more details, please refer to [1] and Appendix A.

## 3. Results and Discussion

In this section, some simulation results are discussed over one regression problem and the MNIST handwritten digit classification problem. For the regression model, performance evaluation goes into the eventual number of hidden nodes required for achieving the expected training error tolerance. For the classification problem, performance evaluation goes into the eventual training and testing accuracy. In this study, the sigmoid activation function g(x) = 1/(1 + exp(-x)) is being used, which is the default in the original SCN algorithms [1].

### 3.1. A Regression Model

The dataset of the regression model was generated by a real-valued function [18]:(8)$$f\left(x\right)=0.2e^{{-(10x-4)}^{2}}+0.5e^{{-(80x-40)}^{2}}+0.3e^{{-(80x-20)}^{2}},$$
where x ∈ [0, 1]. There are 1000 points randomly generated from the uniform distribution [0, 1] in the training dataset. The test set has 300 points generated from a regularly spaced grid on [0, 1]. The input and output attributes are normalized into [0, 1], and all results reported in this research took averages over 1000 independent trials. The settings of parameters are similar to the SCN in [1].

#### 3.1.1. Parameter Tuning for Regression Model

The Materials and Methods section shows that the three heavy-tailed distribution algorithms have user-defined parameters, for example, the power-law index for SCN-Lévy, and location and scale parameters for SCN-Cauchy and SCN-Weibull, respectively. Thus, to illustrate the effect of parameters on the optimization results and to offer reference values to the proposed SCN algorithms, parameter analysis is conducted, and corresponding experiments are performed. Based on the experimental results, for the SCN-Lévy algorithm, the most optimal power law index is 1.1 to achieve the minimum number of hidden nodes (Figure 1). For the SCN-Weibull algorithm, the optimal location parameter α and scale parameter β for the minimum number of hidden nodes are 1.9 and 0.2, respectively (Figure 2). For the SCN-Cauchy algorithm, the optimal location parameter α and scale parameter β for the minimum number of hidden nodes are 0.9 and 0.1, respectively (Figure 3). The values of colorbar in Figure 2 and Figure 3 represent for the mean hidden node number. Lower values mean that the SCN algorithms use less hidden node numbers to reach the training tolerance.

#### 3.1.2. Performance Comparison among SCN, SCN-Lévy, SCN-Cauchy, SCN-Weibull, and SCN-Mixture on the Regression Model

In Table 2, the performance of SCN, SCN-Lévy, SCN-Cauchy, SCN-Weibull, and SCN-Mixture are shown, in which mean values are reported based on 1000 independent trials. In [1], Wang et al. used time cost to evaluate the SCN algorithms performance. In this study, the authors used the mean hidden node numbers to evaluate the performance. The number of hidden nodes associates with the modeling accuracy. Therefore, the authors are interested to analyze if SCN with heavy-tailed distributions can use less hidden nodes to generate high performance, which makes the NNs less complex. According to the experimental results, the SCN-Cauchy used the least number of mean hidden nodes, 59 with an RMSE of 0.0057. The SCN-Weibull has a mean hidden nodes number of 63, with an RMSE of 0.0037. The SCN-Mixture has a mean hidden nodes number of 70, with an RMSE of 0.0020. The mean hidden nodes number for SCN-Lévy is 70. The original SCN model has a mean hidden nodes number of 75. A more detailed training process is shown in Figure 4. With fewer hidden node numbers, the SCN models with heavy-tailed distribution can be faster than the original SCN model. The neural network structure is also less complicated than SCN. Our experimental results on the regression task demonstrate remarkable improvements in modeling performance compared with the current SCN model results. For example, the approximation performance of SCN-Cauchy is shown in Figure 5. The other SCN models with heavy-tailed distributions also have similar performance.

### 3.2. MNIST Handwritten Digit Classification

The handwritten digit dataset contains 4000 training examples and 1000 testing examples, a subset of the MNIST handwritten digit dataset. Each image is a 20 by 20 pixels grayscale image of the digit (Figure 6). Each pixel is represented by a number indicating the grayscale intensity at that location. The 20 by 20 grid of pixels is “unrolled” into a 400-dimensional vector.

#### 3.2.1. Parameter Tuning for Classification Model

Similar to the parameter tuning for the regression model, parameter analysis was conducted to illustrate the impact of parameters on the optimization results and to offer reference values to the MNIST handwritten digit classification SCN algorithms. Corresponding experiments are performed. According to the experimental results, for the SCN-Lévy algorithm, the most optimal power law index is 1.6 to achieve the best RMSE performance (Figure 7). For the SCN-Cauchy algorithm, the optimal location parameter α and scale parameter β for the least RMSE are 0.2 and 0.3, respectively (Figure 8). The values of colorbar in Figure 8 represent for the RMSE.

#### 3.2.2. Performance Comparison among SCN, SCN-Lévy, SCN-Cauchy, and SCN-Mixture on the Classification Model

The performance of the SCN, SCN-Lévy, SCN-Cauchy, and SCN-Mixture are shown in Table 3. Based on the experimental results, both of the SCN-Cauchy, SCN-Lévy and SCN-Mixture have better performance on training and test accuracy, compared with the original SCN model. A detailed training process is shown in (Figure 9). Within around 100 hidden nodes, the SCN models with heavy-tailed distribution perform similarly to the original SCN model. When the number of the hidden node is greater than 100, the SCN models with heavy-tailed distribution have lower RMSE. Since more parameters for weights and biases are initialized in heavy-tailed distribution, this may cause that the SCN with heavy-tailed distributions converges to the optimal values at a faster speed. The experimental results on the MNIST handwritten classification problem demonstrate improvements in modeling performance. It also shows that SCN models with heavy-tail distributions have a better search ability to achieve lower RMSE. The training and test performance of the SCN, SCN-Cauchy, and SCN-Lévy are shown in Figure 10.

## 4. Conclusions

The SCN models have been widely used for regression and classification analysis. Compared with the known randomized learning algorithms for single hidden layer feed-forward neural networks, the SCN randomly assigns the input weights and biases of the hidden nodes in a supervisory mechanism. Since the parameters of the hidden layers are randomly generated in uniform distribution, it might exist optimal randomness in the SCN models. Heavy-tailed distribution has shown optimal randomness in an unknown environment for finding some targets, which plays a significant role in both exploration and exploitation. Therefore, in this research, the authors mainly discussed the effect of different heavy-tailed distributions on the SCN algorithms performance.

In this study, heavy-tailed distributions, such as Lévy distribution, Cauchy distribution, and Weibull distribution, are being used for generating optimal randomness of weights and biases initialization. Experiment results showed improved performance for SCN with heavy-tailed distributions. The experimental results of regression tasks demonstrate remarkable improvements in modeling performance compared with the existing SCN model results. SCN-Mixture, SCN-Lévy, SCN-Cauchy, and SCN-Weibull used less hidden nodes to achieve similar performance with SCN. For the classification model, SCN-Mixture, SCN-Lévy and SCN-Cauchy have higher test accuracy of 91.5%, 91.7% and 92.4%, respectively. Both are higher than the test accuracy of the original SCN. The experimental results showed that SCN models with heavy-tail distributions have a better search ability to achieve better performance.

For future research, the authors will try the heavy-tailed distributions for the DeepSCN models. Since heavy-tailed distributions, such as SCN-Lévy, SCN-Cauchy, and SCN-Weibull, showed improved performance for SCN models. Other heavy-tailed distributions, such as Mittag-Leffler distribution, Pareto-distribution, and more, will be applied for SCN and other algorithms. Randomly assigning a subset of parameters in a learner model is an efficient method for training NNs, which provides an effective solution for machine learning problems. The proposed SCN algorithms with heavy-tailed distributions will be applied for more regression and classification problems in the future.

## Figures and Tables

**Figure 1 entropy-23-00056-f001:**
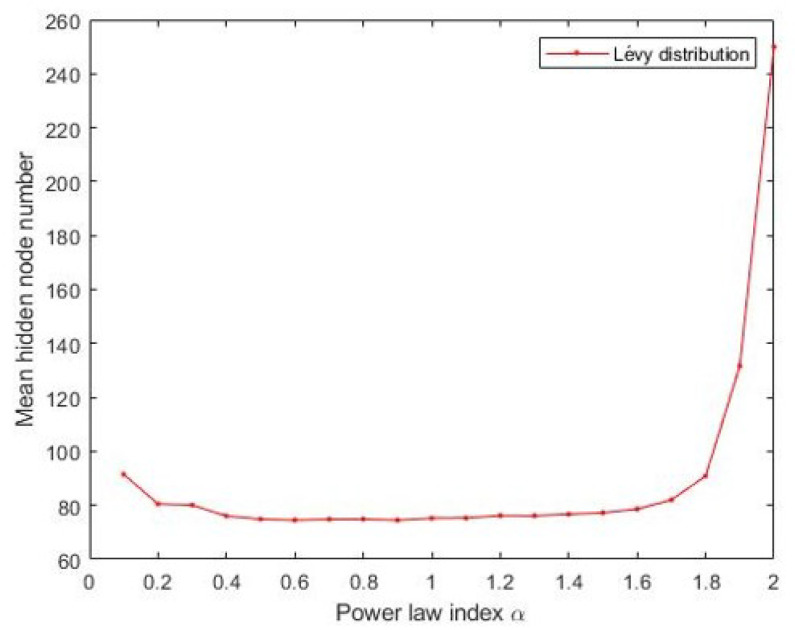
SCN-Lévy parameters tuning for regression model.

**Figure 2 entropy-23-00056-f002:**
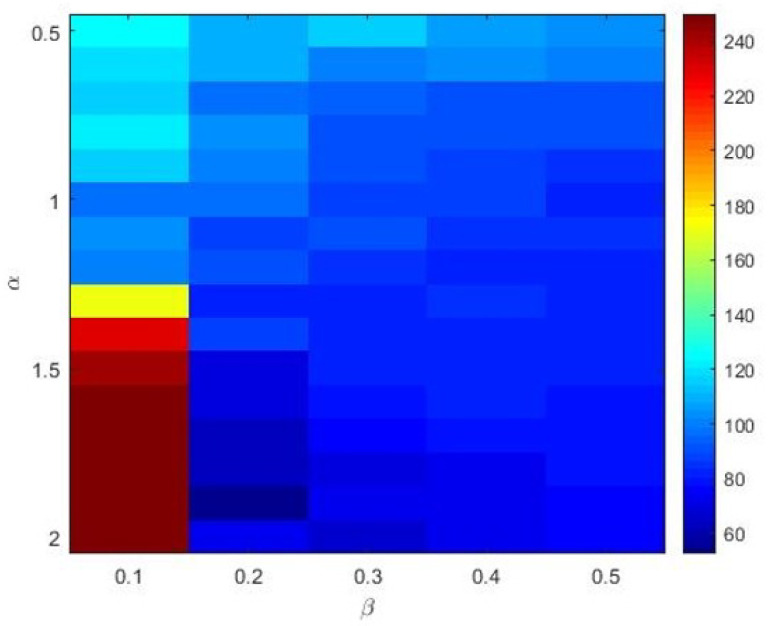
SCN-Weibull parameters tuning for regression model.

**Figure 3 entropy-23-00056-f003:**
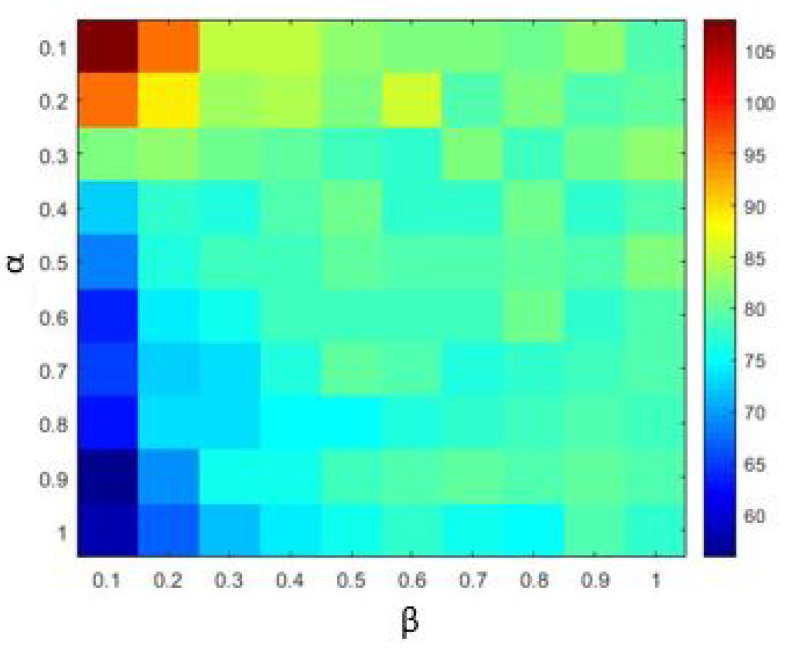
SCN-Cauchy parameters tuning for regression model.

**Figure 4 entropy-23-00056-f004:**
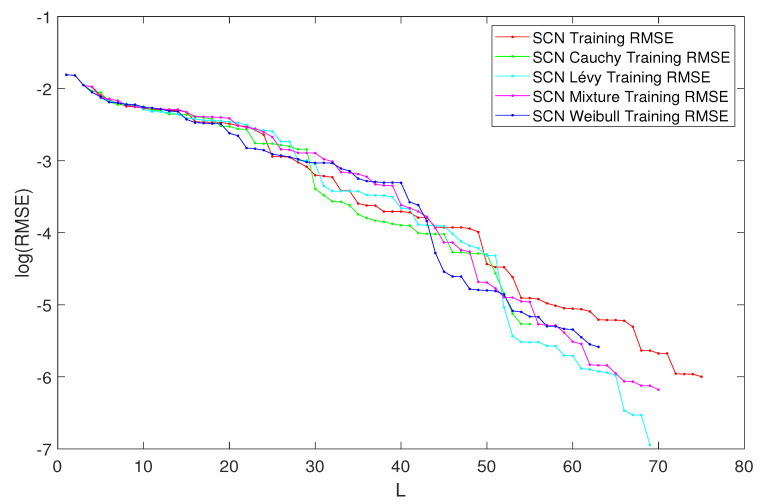
Performances of SCN, SCN-Lévy, SCN-Weibull, SCN-Cauchy, and SCN-Mixture.

**Figure 5 entropy-23-00056-f005:**
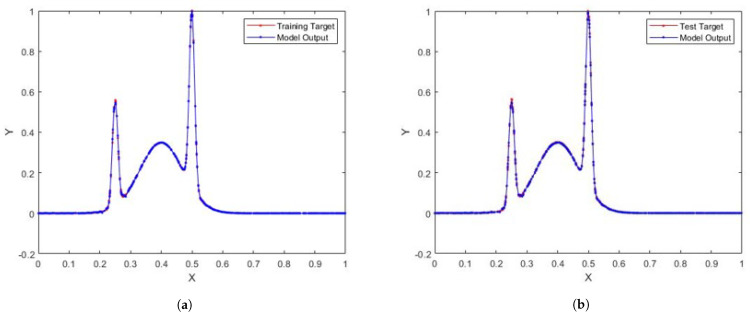
(**a**) SCN-Cauchy training performance for regression model; (**b**) SCN-Cauchy test performance for regression model.

**Figure 6 entropy-23-00056-f006:**
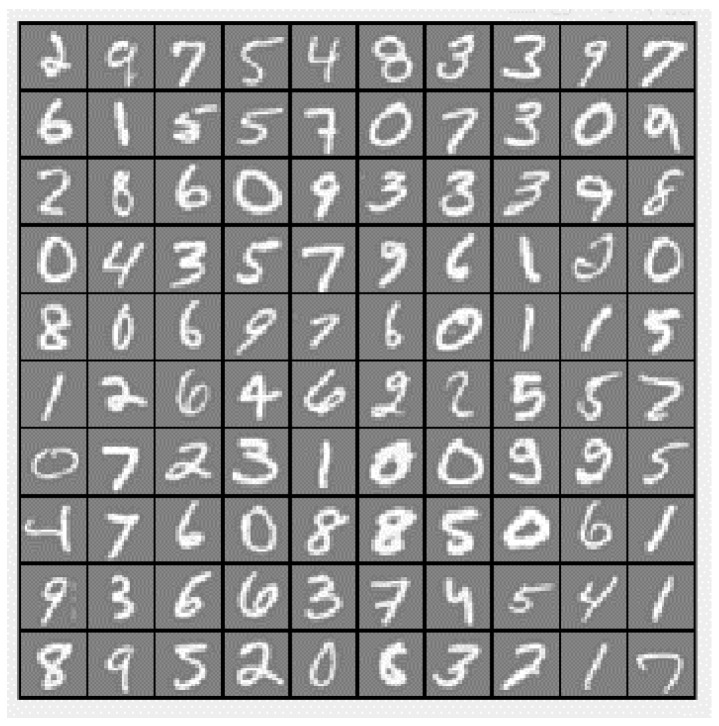
The handwritten digit dataset example.

**Figure 7 entropy-23-00056-f007:**
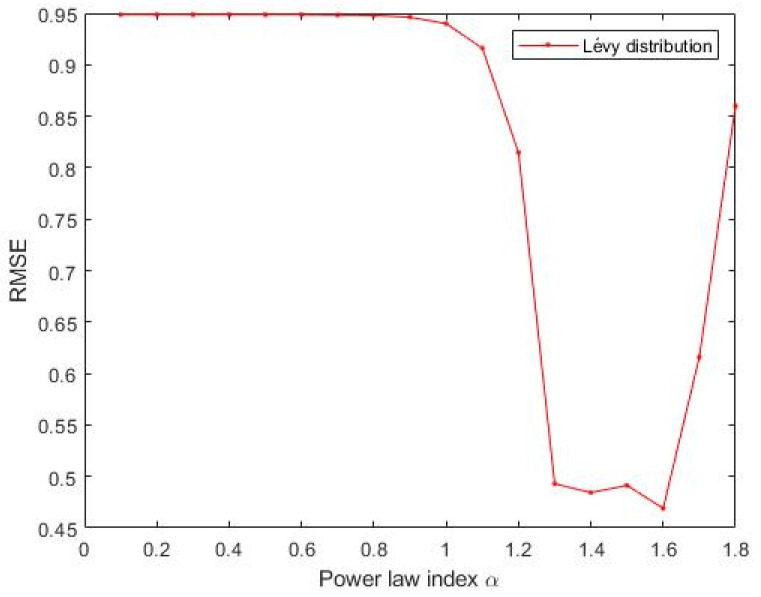
SCN-Lévy parameters tuning for classification model.

**Figure 8 entropy-23-00056-f008:**
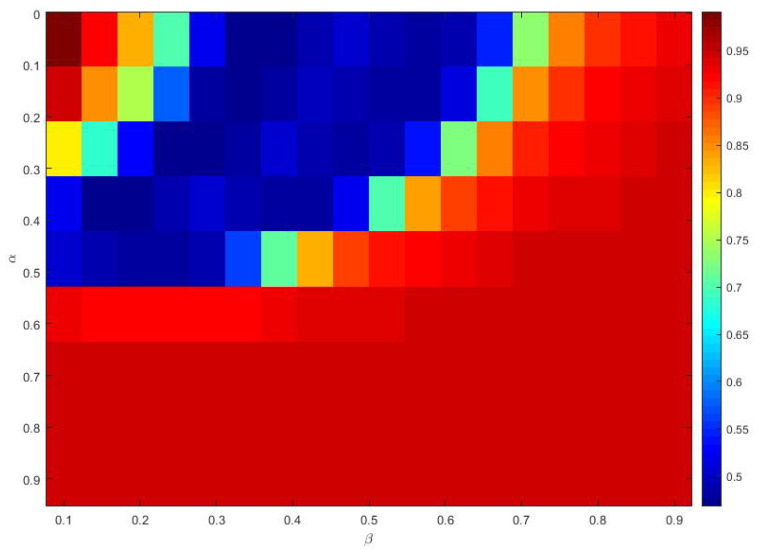
SCN-Cauchy parameters tuning for classification model.

**Figure 9 entropy-23-00056-f009:**
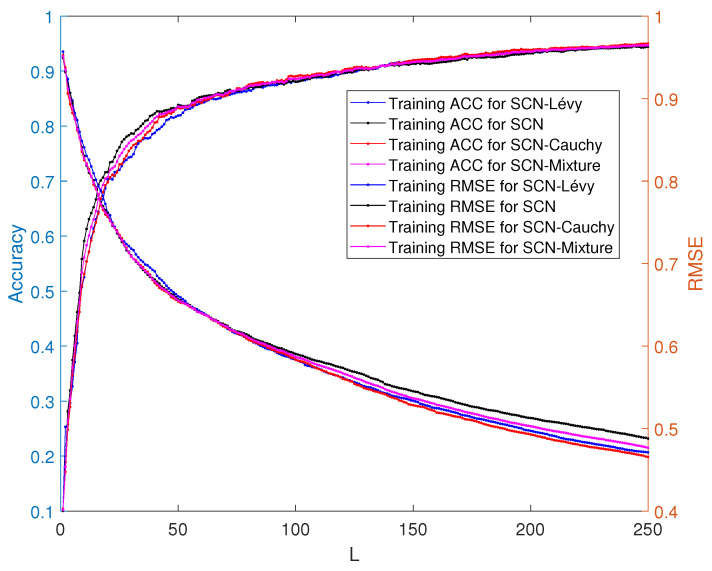
Classification performance of training accuracy between SCN, SCN-Lévy and SCN-Cauchy.

**Figure 10 entropy-23-00056-f010:**
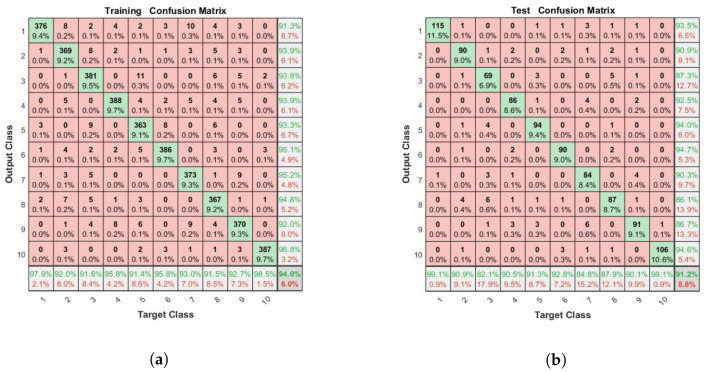
(**a**) SCN training performance for classification model; (**b**) SCN test performance for classification model; (**c**) SCN-Cauchy training performance for classification model; (**d**) SCN-Cauchy test performance for classification model; (**e**) SCN-Lévy training performance for classification model; (**f**) SCN-Lévy test performance for classification model.

**Table 1 entropy-23-00056-t001:** SCNs with key parameters.

Properties	Values
Name:	’Stochastic Configuration Networks’
version:	’1.0 beta’
L:	hidden node number
W:	input weight matrix
b:	hidden layer bias vector
Beta:	output weight vector
r:	regularization parameter
tol:	tolerance
Lambdas:	random weights range
L${}_{max}$:	maximum number of hidden neurons
T${}_{max}$:	maximum times of random configurations
nB:	number of node being added in one loop

**Table 2 entropy-23-00056-t002:** Performance comparison of SCN models on regression problem.

Models	Mean Hidden Node Number	RMSE
SCN	75 ± 5	0.0025,
SCN-Lévy	70 ± 6	0.0010,
SCN-Cauchy	59 ± 3	0.0057,
SCN-Weibull	63 ± 4	0.0037,
SCN-Mixture	70 ± 5	0.0020.

**Table 3 entropy-23-00056-t003:** Performance comparison between SCN, SCN-Lévy and SCN-Cauchy.

Models	Training Accuracy	Test Accuracy
SCN	94.0 ± 1.9%	91.2 ± 6.2%,
SCN-Lévy	94.9 ± 0.8%	91.7 ± 4.5%,
SCN-Cauchy	95.4 ± 1.3%	92.4 ± 5.5%,
SCN-Mixture	94.7 ± 1.1%	91.5 ± 5.3%.

## Data Availability

Not applicable.

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
