# Peer review of "Optimal Randomness for Stochastic Configuration Network (SCN) with Heavy-Tailed Distributions"

_entropy, 2020, doi:10.3390/e23010056_

Round 1

Reviewer 1 Report

The authors propose a stochastic configuration network based on heavy-tailed distribution. This topic was tackled recently by Lu and Ding in an entitled paper "Mixed-Distribution-Based Robust Stochastic Configuration Networks for Prediction Interval Construction." In this paper, the authors considered a heavy-tailed approach using the Laplace distribution and a mixture of Gaussian and Laplace distributions.

The revised paper fails in two directions. The first one is that they present neither how to implement the proposal nor technical details about the heavy-tails model. The second one is that a comparison with the proposal mentioned above is a must.

Consequently, the paper does not provide a significant contribution to this field.

Author Response

Thank you! Please refer to the attached pdf for our responses.

Reviewer 2 Report

The manuscript explores the existence of optimal randomness in Stochastic Configuration Network (SCN). In SCN model, weights and biases are randomly generated in uniform distribution. In this work, authors explore the performance of SCN with choosing three different heave-tail distributions, i.e. Levy, Cauchy and Weibull. The authors claim higher efficiency of SCN with heavy-tail distribution compared to SCN with normal distribution applied by training and testing their model on a linear regression problem, and classification problem is done on MNIST dataset. While the motif of this work is interesting, there are some unclarities regarding the parameters and metrics which make this work irreproducible.     1-It is not clear why authors used a mean number of nodes for parameter tuning of their regression model (section 3.1.1), and used a different metric, RMSE for fine-tuning of their classification model(section 3.2.1).   2- Authors have specified the key parameter in table 1. But some of those parameters are missing from the text and the values are not reported. For instance, what is the chosen regularization parameter? How do they choose the maximum times of random configurations?  Without these type of information how a scientific work can be reproducible?   3-Please justify the usage of the sigmoid function. Have the authors explored other activation functions?  4-The introduction on SCN model is not enough, and having missing information on table 1 parameters, make the manuscript even less understandable for a reader new to SCN. Present the principles of SCN theorem with addressing the key parameters. And be mindful about the notation. Some of these parameters have the same notation as the presented distributions, Lambdas, Betas, etc. 5-In figure 2 and 3, clarify the metric and the colorbars. 6-Use larger numbers and labels for figure 4, 8 and 9. 7-Can authors present the memory cost(RAM) of SCN-Lévy, SCN-Cauchy, and SCN-Weibull and SCN at the optimum found parameters for regression and classification models? 8-"Line 156-158: It also shows that SCN models with heavy-tail distributions have a better search ability to achieve lower RMSE. For example, the training and test performance of the SCN-Cauchy are shown in Fig. 10. " Confusion matrix is given as an example to prove line 156, without presenting normal SCN as the comparison. Either reformat this part or add the normal SCN confusion matrix as well at the bottom of figure 10.

Author Response

(The authors gave the same response as above.)

Reviewer 3 Report

I think the authors report the interesting finding that the performance of the stochastic configuration network for training neural nets improves with heavy tail distributions.

A minor point is the use of the name probability distribution function referring to the probability density function. I know that terminology is not standard, but we should try to.

I recommend that the authors try to explain the reason of the improvement. Is it because large improbable values help the calibration? Is there a way of testing this or other plausible explanation?

Author Response

(The authors gave the same response as above.)

Round 2

Reviewer 1 Report

I read the comments provided by the authors. They do not provide a comparison with the suggested alternative. Consequently, I remain in my decision to reject the paper.

Author Response

Dear editors and reviewers, We really appreciated the encouraging, critical and constructive comments on this manuscript by the reviewers. The comments have been very thorough and useful in improving our manuscript. We strongly believe that the comments and suggestions have increased the value of the revised manuscript. We have taken them fully into account in revision. We are submitting the corrected manuscript with the suggestion incorporated the manuscript for the Round 2. The manuscript has been revised as per the comments given by the reviewer, and our responses to all the comments are as follows: Reviewer #1: 1. I read the comments provided by the authors. They do not provide a comparison with the suggested alternative. Consequently, I remain in my decision to reject the paper. Response- Thank you so much for the comments. In the revised paper, we added the mixture distribution to generate the heavy tailed distributions. The details are described as: In Line 13, we added “Since some mixed distributions show heavy-tailed properties, the mixed Gaussian and Laplace distributions were also studied in this research work.” In Line 17 and 18, we added “For the regression model, SCN-Lévy, SCN-Mixture, SCN-Cauchy, and SCN-Weibull used less hidden nodes to achieve similar performance with SCN. For the classification model, SCN-Mixture, SCN-Lévy, and SCN-Cauchy have higher test accuracy of 91.5%, 91.7% and 92.4%, respectively” From Line 72 to Line 75, we added “To compare with the mixture distributions, the authors also used the mixture distributions for weight and bias generation. A more detailed comparison of the two heavy-tailed methods is shown in Results and Discussion section.” From Line 106 to Line 121, we added the mixture distribution using Gaussian and Laplace distributions. In Line 167 and 168, we changed the title to “Performance comparison among SCN, SCN-Lévy, SCN-Cauchy, SCN-Weibull, and SCN-Mixture on the regression model” In Table 2, we added the results generated from SCN-Mixture. In Figure 4, we added the training performance of the SCN-Mixture. In Line 177, we added “The SCN-Mixture has a mean hidden nodes number of 70, with an RMSE of 0.0020” In Line 198 and 199, we changed the title to “Performance comparison among SCN, SCN-Lévy, SCN-Cauchy, and SCN-Mixture on the classification model” In Table 3, we added the SCN-Mixture’s training and test accuracy on MNIST datasets. In Figure 9, we added the SCN-Mixture training accuracy and RMSE.

Reviewer 2 Report

The authors recognized and accepted the necessary modifications, which is
important to make the presented arguments more convincing to readers. Therefore,
based on the presented updated in the manuscript I recommend the manuscript to
be published in the Entropy.

Author Response

Reviewer #2:

  1. The authors recognized and accepted the necessary modifications, which is
    important to make the presented arguments more convincing to readers. Therefore,
    based on the presented updated in the manuscript I recommend the manuscript to
    be published in the Entropy.

Response- Thank you so much for your comments.